# Mechanistic basis for selective Csm6-2 activation by cyclic penta-adenylate in a type III CRISPR-Cas system

Ruyi Shi [1,4], Mengquan Yang [2,3,4], Yusong Liu [2,3], Haishan Gao [2,3 ✉] & Zhonghui Lin [1 ✉]

## Abstract

Type III CRISPR systems generate cyclic oligoadenylate (cOA, 3 to 6 AMPs) messengers upon detecting viral RNA, activating downstream effectors to defend against viral infection. Although cOA-activated effectors have been extensively characterized, the effectors specific to cA5–one of the most abundant cOA species produced during phage infection–have remained unexplored. Here, we report that the CRISPR ribonuclease Csm6 (Csm6-2) from Actinomyces procaprae selectively employs cA5 as its activator. Csm6-2 utilizes its HEPN domain, rather than the CARF domain, to mediate self-limiting cleavage of cOA activators. Cryo-EM structural analyses reveal that Csm6-2 functions as a homotetramer, and disruption of tetramer formation significantly reduces its ribonuclease activity. Although cA6 and cA5 bind Csm6-2 with comparable affinity, only cA5 induces CARF domain closure, stabilizes the tetramer, and remodels the active site in the HEPN domain. In contrast, the sixth AMP of cA6 imposes significant steric hindrance on CARF domain movement, preventing its closure and subsequent allosteric activation. These findings expand our understanding of the cOA signaling diversity and specific cOA recognition mechanisms in type III CRISPR immunity.

**Keywords** Type III CRISPR; Ribonuclease; Csm6; Cyclic Penta-adenylate; Allosteric Activation

**Subject Categories** Microbiology, Virology & Host Pathogen Interaction; Structural Biology

## Introduction

Bacteria and archaea employ CRISPR-Cas systems to defend against invading foreign genetic elements (Barrangou et al, 2007; Marraffini and Sontheimer, 2008; Mojica and Rodriguez-Valera, 2016). Unlike other CRISPR-Cas systems, type III systems are distinguished by a unique cyclic oligoadenylate (cOA) signaling mechanism, which provides an additional defense to combat viral infection (Athukoralage and White, 2021, 2022; Kazlauskiene et al, 2017; Koonin and Makarova, 2018; Makarova et al, 2015; Molina et al, 2020; Niewoehner et al, 2017; Rouillon et al, 2018). Upon detecting viral mRNA, the Cas10 protein (also referred to as Csm1 or Cmr2) within type III CRISPR-Cas complexes synthesizes ring-shaped cOA molecules, which are composed of three to six AMPs linked by 3'-5' phosphodiester bonds (Athukoralage and White, 2021, 2022; Kazlauskiene et al, 2017; Koonin and Makarova, 2018; Molina et al, 2020; Niewoehner et al, 2017; Rouillon et al, 2018). These messenger cOAs in turn can activate diverse downstream effector proteins, including RNases, DNases, transcription factors, proteases as well as adenosine deaminase (Athukoralage and White, 2022; Baca et al, 2024; Kolesnik et al, 2021; Li et al, 2025; Makarova et al, 2020; Steens et al, 2022).

To date, the most characterized type III CRISPR effectors are the RNA-targeting Csx1/Csm6 proteins. These proteins typically consist of an N-terminal CARF (CRISPR-associated Rossmann Fold) domain and a C-terminal HEPN (higher eukaryotes and prokaryotes nucleotide-binding) domain (Du et al, 2024; Garcia-Doval et al, 2020; Makarova et al, 2020; Molina et al, 2019; Niewoehner and Jinek, 2016; Zhang et al, 2024). When cOA binds to the CARF domain, it allosterically activates ribonuclease activity in the HEPN domain, resulting in nonspecific degradation of ssRNA (Du et al, 2024; McQuarrie et al, 2023; Zhang et al, 2024). To prevent excessive RNA degradation that could cause host cell dormancy or death during early viral infection, cells utilize a group of cOA-degrading enzymes called ring nucleases (Athukoralage et al, 2019; Athukoralage et al, 2020a; Athukoralage et al, 2020b; Athukoralage et al, 2018; Brown et al, 2020; Du et al, 2023; Molina et al, 2022; Molina et al, 2021). Beyond the standalone ring nucleases, certain Csx1/Csm6 proteins, referred to as self-limiting ribonucleases, exhibit intrinsic cOA-degrading activity. This activity is mediated either through their own CARF domains (Athukoralage et al, 2019; Du et al, 2024; Garcia-Doval et al, 2020; Jia et al, 2019; McQuarrie et al, 2023; Smalakyte et al, 2020) or via the integration of a viral anti-CRISPR ring nuclease (Samolygo et al, 2020; Zhang et al, 2024), or both CARF and HEPN domains (Jia et al, 2019; Smalakyte et al, 2020), providing an off-switch to downregulate cOA signaling.

Although the Cas10 nucleotidyl cyclase generates a spectrum of cOAs (including cA$_3$, cA$_4$, cA$_5$, and cA$_6$), most characterized type

[1] College of Chemistry, Fuzhou University, Fuzhou, China. [2] Westlake Laboratory of Life Sciences and Biomedicine, Hangzhou, China. [3] School of Life Sciences, Westlake University, Hangzhou, China. [4] These authors contributed equally: Ruyi Shi, Mengquan Yang. ✉E-mail: gaohaishan@westlake.edu.cn; zhonghui.lin@fzu.edu.cn

III effectors prefer $cA_4$ as their cognate ligand, with a subset utilizing $cA_6$ (Athukoralage and White, 2022; Kolesnik et al, 2021; Steens et al, 2022). This preference is likely attributed to the homodimeric assembly of their CARF domains, which favor recognition of ligands with twofold symmetry. Likewise, the CBASS (cyclic oligonucleotide-based anti-phage signaling system) effector NucC recognizes $cA_3$ as a homotrimer (Lau et al, 2020). In contrast, the distant CARF homologs, SAVED (SMODS-associated and fused to various effector domains) domains function as monomers and bind a range of cyclic di- and trinucleotides (Athukoralage and White, 2022). For example, the SAVED domain of Cap4 comprises two tandem CARF-like subdomains and specifically recognizes $cA_3$ as its activator (Lowey et al, 2020), and the CRISPR-associated Lon protease (CalpL) contains two pseudosymmetric CARF-like domains that binds a $cA_4$ activator (Rouillon et al, 2023).

Previous bioinformatic analysis identified a Csm6 homolog from *Actinomyces procaprae*, termed Csm6-2, which consists of two tandem CARF-HEPN modules in a single polypeptide chain (Hoikkala et al, 2024). Csm6-2 is present in 16 type III-D loci and is hypothesized to have originated from a Csm6 ancestor via gene fusion (Hoikkala et al, 2024), thereby representing an uncharacterized class of cOA-responsive CARF effectors. In the present study, we show that Csm6-2 is preferentially activated by $cA_5$. The cryo-EM structures of Csm6-2 and its complexes with $cA_5$ and $cA_6$ reveal that Csm6-2 functions as a homotetramer, and the tetramerization is critical for effective ribonuclease activity. Further structural analyses reveal the molecular basis for $cA_5$-specific recognition and the allosteric mechanism that activates ssRNA cleavage. These findings advance our understanding of signaling molecules diversity and regulatory mechanisms in type III CRISPR systems.

# Results

## Csm6-2 is preferentially activated by $cA_5$

Csm6-2 was previously shown to be activated by $cA_6$ but not $cA_3$ or $cA_4$ (Hoikkala et al, 2024). We characterized Csm6-2's ribonuclease activity in response to various cyclic oligoadenylates including $cA_5$, using a gel-based ssRNA cleavage assay. At a saturating concentration of cOAs (100 nM), $cA_5$ triggered robust ssRNA cleavage with only 1 nM Csm6-2 (Fig. 1A). In contrast, $cA_6$ induced only weak cleavage activity, even at a high enzyme concentration of 100 nM. No detectable cleavage was observed with $cA_4$ under these conditions. To further characterize the cOA activation profiles, we employed RNA activators which consist of a poly (A) stretch followed by a protective poly (U) on the 3' side (Fig. 1B). The protecting poly (U) can be cleaved by a uracil-preferring *Lbu*Cas13a upon target RNA detection, thereby releasing a linear oligoadenylate activator with a 2′,3′-cyclic phosphate end ($A_n$>P) (Gootenberg et al, 2018). These $A_n > P$ activators can stimulate Csm6 ribonuclease activity similarly as their cyclic counterparts (Gootenberg et al, 2018; Liu et al, 2021). Consistent with the gel-based results, $A_5 > P$ demonstrated stronger activation of Csm6-2 than $A_6 > P$ (Fig. 1B). In contrast, $A_2 > P$, $A_3 > P$, and $A_4 > P$ elicit no detectable activity. Surface plasmon resonance (SPR)-based

ligand binding assay demonstrated that $cA_5$ and $cA_6$ bind to Csm6-2 with comparable affinity, exhibiting $K_D$ values of 1.41 nM and 3.72 nM, respectively (Fig. 1C; Appendix Fig. S1A), indicating that the dramatic difference in their activation potency may not be attributed to differences in binding affinity.

Together, these findings establish $cA_5$ as the preferential activator of Csm6-2, distinguishing it from other characterized Csm6 proteins that rely on $cA_4$ or $cA_6$.

## Csm6-2 functions as a self-limiting ribonuclease by degrading cOA activators through its HEPN domain

As most Csm6 proteins function as self-limiting ribonucleases through intrinsic cOA degradation, we sought to determine whether Csm6-2 shares this capability. HPLC and MALDI-TOF MS analyses revealed that Csm6-2 could degrade both $cA_5$ and $cA_6$, generating predominantly $A_2 > P$ and $A_3 > P$ products (Fig. 1D; Appendix Fig. S1B–D). Such activity was completely abolished by a point mutation in HEPN domain (R723A) (Fig. 1D; Appendix Fig. S1B), suggesting that Csm6-2 degrades cOAs through its HEPN domain. Furthermore, pre-incubation of $cA_5$ with wild-type but not mutant Csm6-2, greatly diminished $cA_5$'s activation on ssRNA cleavage (Fig. 1E). These results thereby suggest that Csm6-2 self-limits its ribonuclease activity by degrading cOAs through its HEPN domain.

## Overall architecture of Csm6-2

We next determined the cryo-EM structure of Csm6-2 in its apo state at a resolution of 2.59 Å (Appendix Fig. S2; Table 1). The structure shows that the two tandem CARF-HEPN repeats, CARF1-HEPN1 (residues 1–400) and CARF2-HEPN2 (residues 401–797), interwine to form a intramolecular dimer (Fig. 2A,B). Four of these molecules further assemble into an homotetramer, primarily through contacts mediated by the CARF2 domains (Fig. 2C–E). First, two monomers (A/B or C/D) dimerize through a pair of antiparallel β-strands involving residues V466, L473 and I488 (Fig. 2F). Two such dimers then associate to form the tetramer, stabilized largely by hydrophobic interactions between four adjacent α-helices, with key contributions from L473 and L474 (Fig. 2G). Mutation of L473 and L474 to glutamates disrupted tetramer formation, resulting in dimeric assembly (Fig. 2H,I) and a significant reduction in ssRNA cleavage activity (Fig. 2J), suggesting that tetramerization is critical for effective ribonuclease activity of Csm6-2.

## Structure of Csm6-2 in complex with cOA activators

To illuminate the structural basis for cOA recognition by Csm6-2, we next determined the structure of Csm6-2 in complex with either $cA_5$ or $cA_6$, as both ligands bind to Csm6-2 with high affinity. The structure of Csm6-2 R723A-$cA_6$ and Csm6-2 H369A-$cA_5$ complexes were solved at 2.53 Å and 2.67 Å, respectively (Appendix Fig. S2; Table 1). Both structures reveal a homotetrameric assembly consist with the apo form (Fig. 3A; Appendix Fig. S3A). Each monomer binds a single cOA within a composite pocket formed by the CARF1 and CARF2 domains (Fig. 3B–E). The cOA molecules

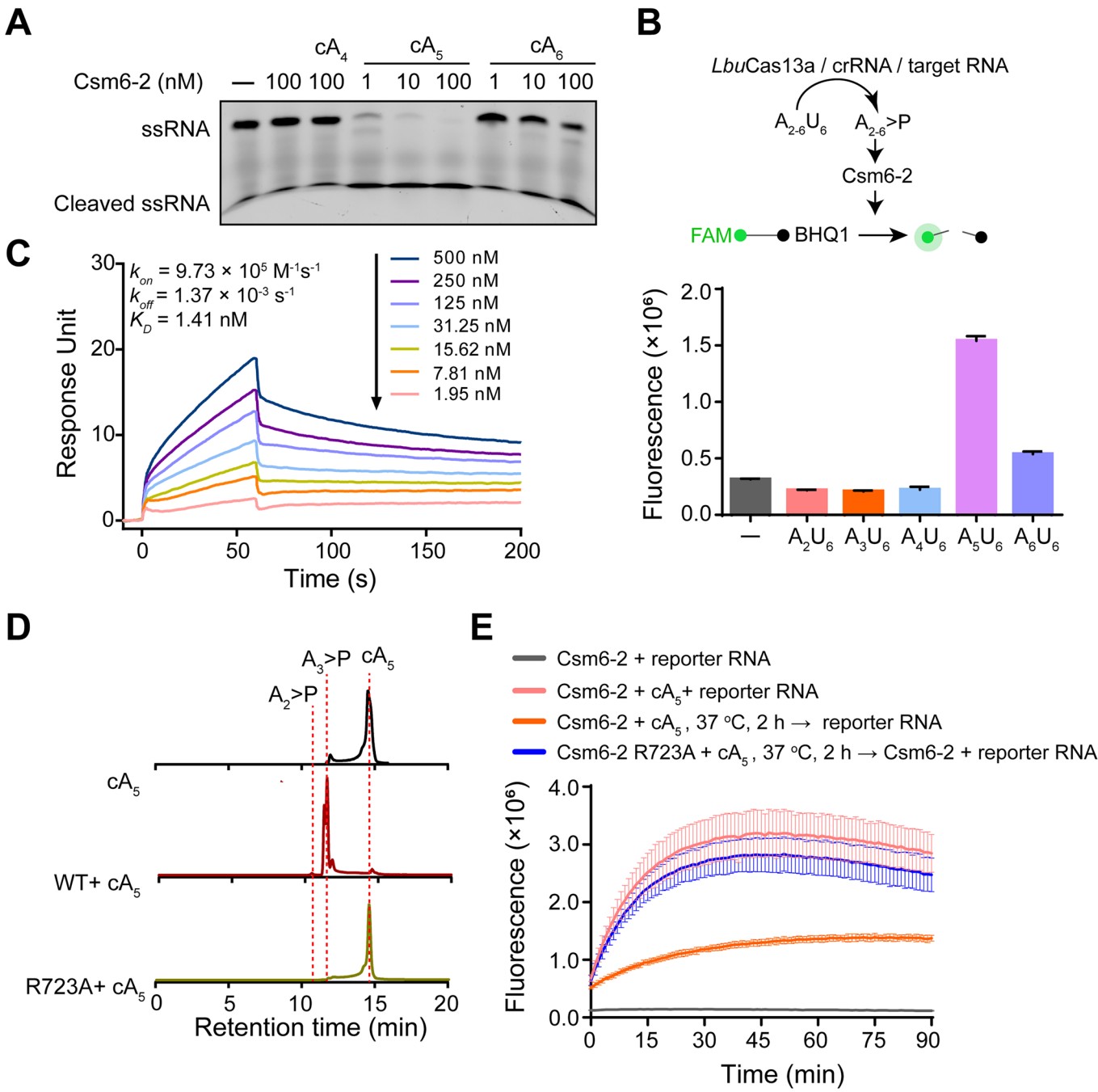

**Figure 1.  Csm6-2 is preferentially activated by cA₅.**

(A) Effect of various cOAs on the stimulation of ssRNA cleavage by Csm6-2. 1~100 nM Csm6-2 protein was incubated with 100 nM cOAs and 250 nM FAM-labeled ssRNA at 37 °C for 45 min, followed by denatured polyacrylamide gel analysis. (B) *Lbu*Cas13a-Csm6-2 tandem assay assessing linear oligonucleotide activation. Top: Assay schematic. Bottom: FRET-based cleavage with 40 nM *Lbu*Cas13a, 20 nM crRNA, 100 pM target RNA, 100 nM Csm6-2, 200 nM reporter RNA, and 2 μM A₂₋₆U₆ activator. Values are means ± SD, n = 3 replicates. (C) Dose–response curves from SPR measurements at indicated concentrations of cA₅ against immobilized Csm6-2 protein. (D) HPLC analysis of cA₅ degradation by WT and mutant Csm6-2. Reactions were conducted at 37 °C for 2 h, using 40 μM cA₅ and 2 μM Csm6-2. (E) FRET-based ssRNA cleavage assay showing reduced ribonuclease activity after pre-incubation of 1 nM Csm6-2 with 10 nM cA₅. Values are means ± SD, n = 3 replicates. Source data are available online for this figure.

**Table 1. Cryo-EM data collection, refinement, and validation statistics.**

| Complexes | Apo ApCsm6 | ApCsm6-cA$_6$ | ApCsm6-cA$_5$ |
|---|---|---|---|
| PDB ID | 9W3U | 9W3V | 9W3W |
| EMDB ID | EMD-65609 | EMD-65610 | EMD-65611 |
| **Data collection and processing** | | | |
| Microscope | FEI Titan Krios | | |
| Voltage | 300 | | |
| Electron dose (e$^-$/Å$^2$) | 50 | | |
| Defocus range (μm) | −1.0 to −2.0 | | |
| Detector | Gatan K3 | Falcon 4i | Falcon 4i |
| Magnification | 81,000 | 130,000 | 130,000 |
| Pixel size (Å/pixel) | 1.087 | 0.97 | 0.97 |
| Micrographs (no.) | 1608 | 5340 | 1994 |
| Final particles (no.) | 149,420 | 667,978 | 143,784 |
| Symmetry imposed | D2 | D2 | D2 |
| Map resolution (Å) | 2.59 | 2.53 | 2.67 |
| FSC threshold | 0.143 | 0.143 | 0.143 |
| **Model composition** | | | |
| Chains | 4 | 8 | 8 |
| Atoms | 22,972 | 22,636 | 24,048 |
| Protein residues | 3044 | 2928 | 3132 |
| Nucleotide | 0 | 24 | 20 |
| Ligands | 0 | 0 | 0 |
| **Refinement** | | | |
| Initial model used | Alphafold 2 | | |
| Model resolution (Å) | 2.8 | 2.8 | 3.0 |
| FSC threshold | 0.143 | 0.143 | 0.143 |
| Map sharpening B factor (Å$^2$) | −106.9 | −101.7 | −94.4 |
| **Validation** | | | |
| MolProbility score | 2.40 | 1.77 | 1.86 |
| Clash score | 7.05 | 3.88 | 4.75 |
| Rotamer outliers (%) | 4.39 | 2.24 | 2.30 |
| C$_\beta$ outliers (%) | 0.36 | 0.00 | 0.04 |
| **R.m.s deviations** | | | |
| Bonds length (Å) | 0.005 | 0.004 | 0.003 |
| Bonds Angle (°) | 0.879 | 0.727 | 0.682 |
| **Ramachandran plot (%)** | | | |
| Favored | 90.73 | 95.24 | 95.11 |
| Allowed | 8.13 | 3.64 | 4.18 |
| Outliers | 1.14 | 1.12 | 0.71 |

exhibit well-resolved electron density except for one AMP of cA$_6$ (AMP-6), which is poorly-defined (Fig. 3C,E). The phosphodiester backbone of the cOA binds atop the interface between CARF-1 and CARF-2 domains, with two adenine groups (from AMP-3 and -4) coordinated by CARF2 and the remaining bases recognized by CARF1 (Fig. 3B,D).

## Mechanism of specific cOA recognition

The overall structure of cA$_6$-bound Csm6-2 closely resembles that of its apo form, exhibiting only minor conformational changes (Fig. 4A; Movie EV1). In contrast, cA$_5$ binding triggers a pronounced conformational shift in the CARF domain, which transitions from an open to a closed state, encapsulating the entire cA$_5$ molecule (Figs. 3B and 4B,C; Movie EV2).

In both cA$_5$- and cA$_6$-bound structures, residues K533 and T175 project their side chains into the cOA ring center, coordinating the 5'-phosphates of AMP-3 and AMP-5, respectively (Fig. 4D,E). Alanine substitution of T175, K533 or both residues abolished the ability of both cA$_5$ and cA$_6$ to activate Csm6-2 ribonuclease activity (Fig. 4F; Appendix Fig. S3B). The adenine moiety of AMP-1 is coordinated by backbone carbonyl oxygens of D100 and G101, and the adenine group of AMP-3 forms hydrogen bonds with S456 and Q532 (Fig. 4D,E). Similarly, the adenine group of AMP-4 interacts with R422 and L530. The 5'-phosphate group of AMP-2 is stabilized by backbone amide nitrogens from E10 and G11, while the phosphate groups of AMP-1 and AMP-4 are primarily coordinated by the guanidine side chain of R415 (Fig. 4D,E).

Notable differences between cA$_5$ and cA$_6$ binding occur at AMP-2 and AMP-5. The adenine group of AMP-2 in cA$_5$ forms a π-π stacking with F68, whereas the corresponding group in cA$_6$ is sandwiched between L18 and L171, and the segment containing F68 is structurally disordered (Fig. 4D,E). In addition, the adenine group of AMP-5 in cA$_5$ binds within a compact pocket involving residues L141, V560, L561, I566 and D100, while the corresponding group in cA$_6$ is stabilized primarily by base-stacking interactions with AMP-1 (Fig. 4D,E). Furthermore, the adenine moiety of AMP-6 in cA$_6$ makes minimal contact with the protein and appears highly flexible (Fig. 4E).

Compared to apo Csm6-2, cA$_5$ binding induces a marked rearrangement of the cOA binding site, particularly in the CARF1 domain (Fig. 4G). Of note, R25 on the CARF1 lid helix shifts by 8.5 Å to coordinate the 5'-phosphate of AMP-1 (Fig. 4G; Movie EV3). However, such movement does not occur upon cA$_6$ binding, due to a steric clash between R25 and the adenine group of AMP-6 (Fig. 4H).

Together, these structural observations elucidate the molecular basis for Csm6-2's preferential activation by cA$_5$ over cA$_6$.

## Mechanism of allosteric activation

Compared to the apo structure, cA$_5$ binding induces the formation of an additional tetramerization interface in the CARF domain (Fig. 5A,B). In particular, the V560–G570 loop, which is disordered in the apo state, becomes well-ordered upon cA$_5$ binding and establishes stable interactions with its counterpart in adjacent monomers (Fig. 5A,B). As a result, the interfacial area between monomers increases from 428.8 Å$^2$ to 651.1 Å$^2$. In contrast, the tetramerization interface in the cA$_6$-bound state remains nearly unchanged from the apo form (Fig. 5C). Since tetramerization is critical for Csm6-2's ribonuclease activity, these additional inter-monomer contacts upon cA$_5$ binding may stabilize the catalytically active tetramer.

As described above, cA$_5$ binding triggers a substantial conformational change in the CARF domain (Fig. 4C; Movie EV3). This transition propagates to the HEPN domain, reorganizing the

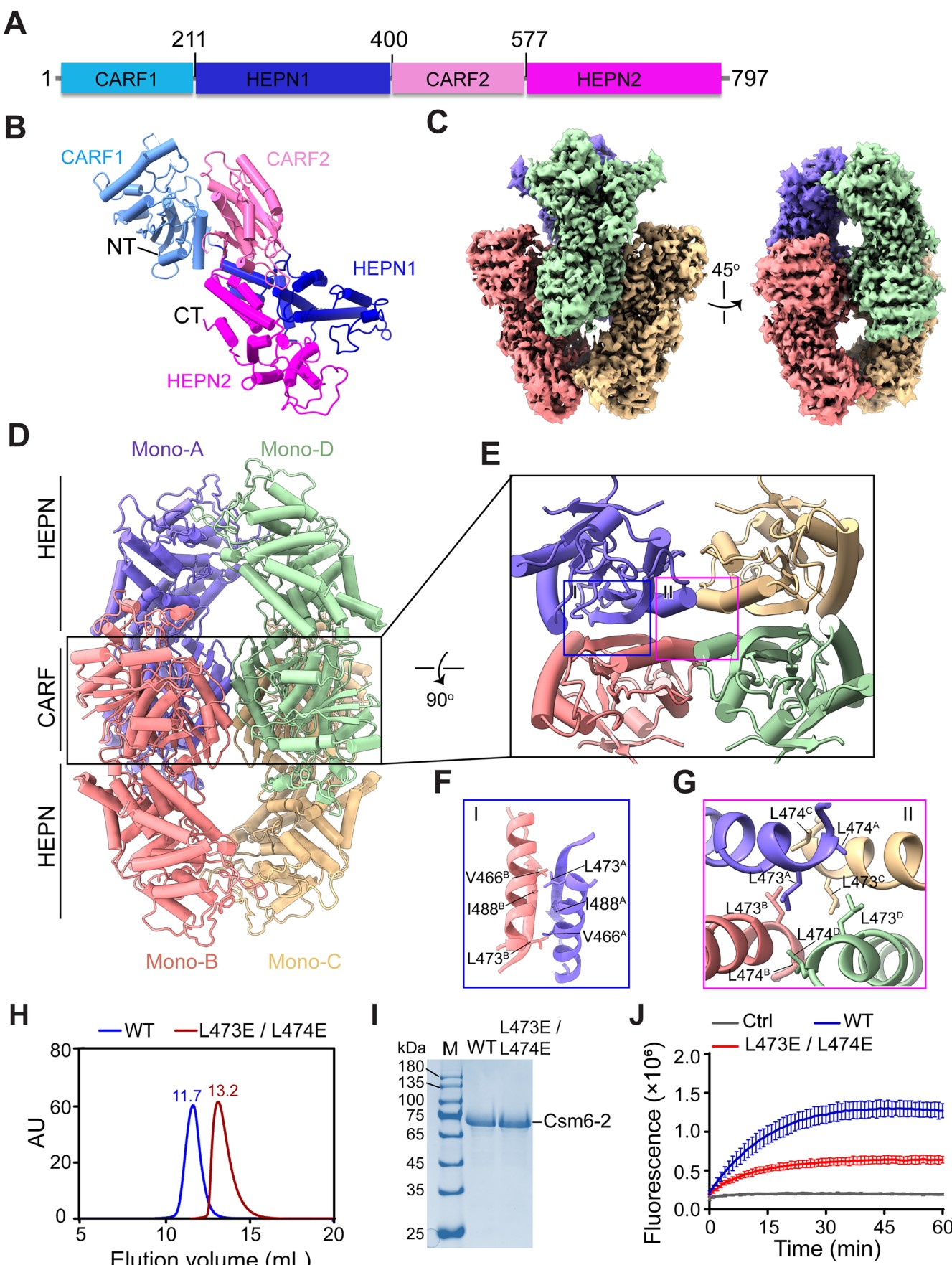

◀  **Figure 2.  Cryo-EM structure of Csm6-2.**

(A) Schematic diagram of the domain architecture of Csm6-2. (B) Structure representation of an Csm6-2 monomer. Subdomains are colored as in (A). (C) Cryo-EM map for Csm6-2, with each monomer shown in a distinct color. (D) Cartoon representation of Csm6-2. (E-G) Close-up views of the tetramerization interfaces mediated by the CARF2 domains. (H) Size-exclusion chromatography profiles of WT and L473E/L474E mutant proteins. (I) Coomassie blue-stained gel of recombinant Csm6-2 proteins. (J) Ribonuclease activity of WT and mutant Csm6-2, measured using a FRET-based ssRNA cleavage assay. 5 nM Csm6-2 and 5 nM $cA_5$ were incubated with 200 nM FAM/BHQ1-labeled ssRNA at 37 °C. Fluorescence intensities were recorded at 1-min intervals. Values are means ± SD, $n = 3$ replicates. Source data are available online for this figure.

ribonuclease catalytic center (Fig. 5D) and altering the positions of key catalytic residues including R295, H369, R723, and H730, all of which are essential for ssRNA cleavage (Fig. 5F; Appendix Fig. S3C). In particular, the guanidinium side chain of R295 rotates by approximately 90° upon $cA_5$ binding, whereas in the apo state it is stabilized by interacting with E299 (Fig. 5D). This mechanism resembles the activation of *Tt*Csm6 by $cA_4$, in which ligand binding releases an catalytic arginine (R415) from a glutamate residue (E332), thereby positioning its guanidine group in the catalytic center (Du et al, 2024). In comparison, the conformational changes induced by $cA_6$ binding are relatively smaller and closely resemble the apo conformation (Fig. 5E).

Collectively, these results reveal a dual function for $cA_5$ in orchestrating both global tetramer stabilization and local active-site remodeling, which together enable full activation of Csm6-2 ribonuclease activity.

## Discussion

Type III CRISPR-Cas systems feature the signature Cas10 protein that generates cOA messengers (typically containing 3–6 AMPs) in response to viral mRNA detection (Makarova et al, 2015). While numerous cOA-dependent effector proteins have been characterized, it is striking that no $cA_5$-dependent effectors have been identified to date, particularly given that $cA_5$ has been shown to be one the most abundant cOA species produced during phage infection (Smalakyte et al, 2020). In this study, we report that Csm6-2 from the type III CRISPR system of *Actinomyces procaprae* preferentially utilizes $cA_5$ as its activator.

Most characterized Csm6 proteins assemble composite CARF domains through homodimerization and preferentially bind symmetric ligands like $cA_4$ and $cA_6$ (Athukoralage and White, 2022). Unlike canonical Csm6 proteins, Csm6-2 contains two tandem CARF-HEPN modules (CARF1-HEPN1-CARF2-HEPN2) within a single polypeptide chain, forming an asymmetric binding pocket. This unique architecture accommodates $cA_5$ perfectly, with three AMPs bound to CARF-1 and two to CARF-2. Although $cA_6$ can bind, only five of its six AMPs are unambiguously positioned in the CARF domain. The sixth AMP exhibits weak electron density and minimal protein interactions. Consistent with this structural preference for $cA_5$, at a saturating cOA concentration, $cA_5$ triggers robust ssRNA cleavage with as little as 1 nM Csm6-2, whereas $cA_6$ elicits only weak activity even at 100 nM enzyme.

Interestingly, SPR-based binding assay demonstrated that $cA_5$ and $cA_6$ bind Csm6-2 with comparable affinity. This suggests that the marked difference in their ability to activate ribonuclease activity cannot be attributed primarily to differences in binding affinity. Compared to the structure of apo Csm6-2, $cA_5$ binding induces a large structural movement of CARF-1 toward CARF-2, converting the CARF domain to a closed conformation (Movie EV2). The CARF domain conformational changes propagate to the HEPN domain, reorganizing the catalytic R-$X_{4.6}$-H motif into an active configuration that enables ribonuclease activity. In contrast, $cA_6$ binding cannot induce full CARF domain closure due to steric interference between the CARF-1 lid helix and $cA_6$'s sixth AMP, explaining its significant weaker activation potency relative to $cA_5$. In addition to inducing conformational changes within individual Csm6-2 monomers, $cA_5$ but not $cA_6$ binding also enhances tetramerization by significantly expanding the oligomerization interface (~223 Å² increase versus apo state). Since tetramer formation is critical for Csm6-2's ribonuclease activity, this enhanced interface therefore stabilizes the catalytically active tetramer.

To prevent excessive RNA cleavage that could induce host cell dormancy or death during early viral infection, many Csm6 proteins act as self-limiting ribonucleases via intrinsic cOA-degrading activity. This activity is typically mediated by their CARF domains (Athukoralage et al, 2019; Du et al, 2024; Garcia-Doval et al, 2020; Jia et al, 2019; McQuarrie et al, 2023; Smalakyte et al, 2020) or by the domain fusion of a standalone ring nuclease (Samolygo et al, 2020; Zhang et al, 2024). In contrast, the CARF domain of Csm6-2 lacks cOA-cleaving activity, instead, it relies on its HEPN domain to degrade excess cOA. As a consequence, cOA bound within the CARF domain may be protected from degradation, consistent with our observation that pre-incubation of cOA with Csm6-2 did not completely eliminate Csm6-2's ribonuclease activity (Fig. 1E). The mechanism by which host cells ultimately terminate Csm6-2-cOA signaling is currently unclear. However, it is possible that cells may need to maintain basal-level immunity to provide continuous protection against persistent environmental threats. Alternatively, it may be a cellular response to avoid viral ring nucleases. For example, *Bacteroides fragilis* type III CRISPR-Cas system produces S-adenosyl methionine (SAM)-adenosine monophosphates (AMPs) rather than cOA signalling for antiviral defence (Chi et al, 2023). Future cell-based studies should address this critical regulatory gap in type III CRISPR-mediated immunity.

In summary, this study establishes Csm6-2 as the first $cA_5$-specific ribonuclease effector in type III CRISPR systems. Structural and biochemical analyses reveal that its unique tandem CARF-HEPN architecture forms an asymmetric binding pocket optimized for $cA_5$ recognition. $cA_5$ binding allosterically activates robust RNA cleavage by promoting both global tetramer stabilization and local active-site remodeling. These findings expand our understanding of the functional diversity of cyclic oligonucleotide signaling in prokaryotic antiviral immunity.

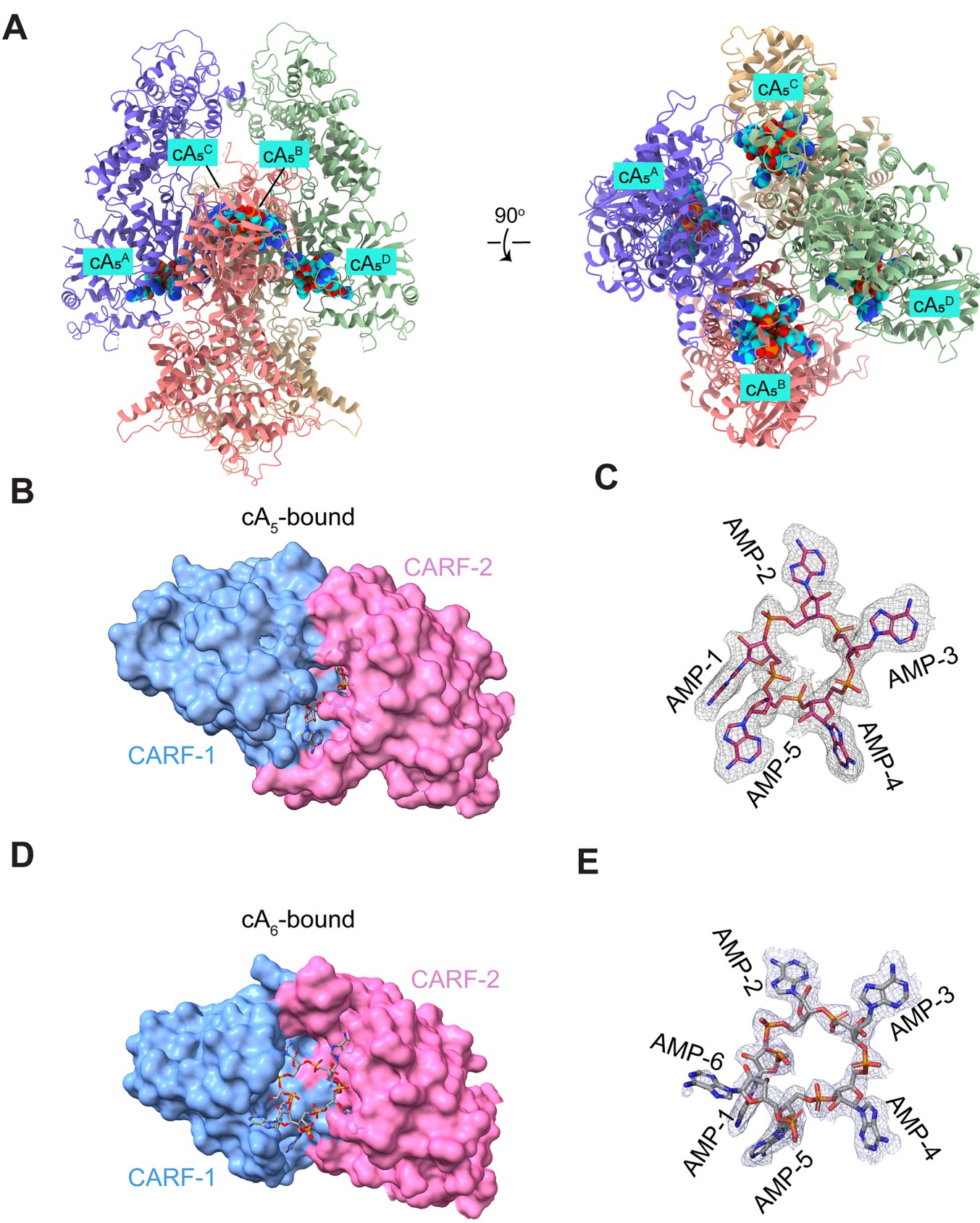

◀ **Figure 3. Structure of Csm6-2 in complex with cOA.**

(A) Cartoon diagrams of Csm6-2-cA$_5$ complex, with cA$_5$ molecules shown as sphere. (B) Surface representation of the Csm6-2 CARF domain bound to cA$_5$. The CARF-1 and CARF-2 moieties are colored in blue and pink, respectively. (C) Stick representation of cA$_5$ overlaid with its cryo-EM map. (D) Surface representation of the Csm6-2 CARF domain bound to cA$_6$. (E) Stick representation of cA$_6$ overlaid with its cryo-EM map.

# Methods

### Reagents and tools table

| Reagent/resource | Reference or source | Identifier or catalog number |
|---|---|---|
| **Experimental models** | | |
| *E. coli* Rosetta (DE3) | This study | |
| **Recombinant DNA** | | |
| pET-28a-*Ap*Csm6-2 | GenScript, Nanjing, China | Customized |
| **Antibodies** | | |
| **Oligonucleotides and other sequence-based reagents** | | |
| cA$_4$ | Biolog Life Science Institute, Germany | Cat # C335 |
| cA$_5$ | Biolog Life Science Institute, Germany | Cat # C394 |
| cA$_6$ | Biolog Life Science Institute, Germany | Cat # C332 |
| 5'-FAM-ACUGCAACGCAAUAUACCAUAGCU-3' | Sunya, Fuzhou, China | Customized |
| 5'-FAM-UGUUCGACGA-BHQ1-3' | Sunya, Fuzhou, China | Customized |
| 5'-FAM-10A-BHQ1-3' | Sunya, Fuzhou, China | Customized |
| 5'-A$_{2-6}$U$_6$-3' | Sunya, Fuzhou, China | Customized |
| R295A-F | This study | GGCCTGGCGGTTGCAGCATGGGTCGAAAAACACTATGAGGCG |
| R295A-R | This study | GTGTTTTTCGACCCATGCTGCAACCGCCAGGCCAGCACCA |
| H369A-F | This study | TGCCGTTGCTGATGCTGCTGCGCCGACCGCGAG |
| H369A-R | This study | CGCAGCAGCATCAGCAACGGCACGGTTGCCAACCGGAC |
| R723A-F | This study | GTTGTACGAGGTTGCCAACGAAGTGCGGTTGACCCACGGTG |
| R723A-R | This study | GGGTCAACCGCACTTCGTTGGCAACCTCGTACAACAGTCTCAGCAG |
| H730A-F | This study | GTGCGGTTGACCGCCGGTGATAGCTCGGTTGACGAGGCGG |
| H730A-R | This study | AACCGAGCTATCACCGGCGGTCAACCGCACTTCGTTGCGAAC |
| T175A-F | This study | GGTGGAGCGGCCATGATGTGCCTGTCCGCT |
| T175A-R | This study | CATCATGGCCGCTCCACCGATCATGCTAACGAC |
| K533A-F | This study | GGGCCAAGCGGGCGCAGTCATCGGCGCGCTGG |
| K533A-R | This study | GACTGCGCCCGCTTGGCCCAGGCCGACTACGGC |
| L473E / L474E-F | This study | GCATCGGTGGAAGAGACCACCCCGAAAGAT |
| L473E / L474E-R | This study | GGTGGTCTCTTCCACCGATGCACAGGTTTTTAC |
| **Chemicals, enzymes and other reagents** | | |
| IPTG | LabLead, China | Cat # 0487072851 |
| Ni-NTA | Union Biotech, China | Cat # USNi250 |
| Source 15Q | GE Healthcare Life Sciences, USA | Cat # 17094720 |
| Superdex 200 10/300 GL | GE Healthcare Life Sciences, USA | Cat # 17517501 |
| Trifluoroacetic acid | Energy Chemical, China | Cat # W810031 |
| Acetonitrile | Fisher Chemical, USA | Cat # F22M6E202 |
| **Software** | | |
| Biacore Insight Evaluation Software | Cytiva, USA | https://www.cytivalifesciences.com/support/software/biacore-downloads/biacore-insight-evaluation-software |
| MotionCor2 | Zheng et al, 2017 | https://emcore.ucsf.edu/ucsf-software |
| cryoSPARC | Punjani et al, 2017 | https://cryosparc.com/ |
| UCSF Chimera | Pettersen et al, 2004 | https://www.cgl.ucsf.edu/chimera/ |
| COOT | Emsley et al, 2010 | https://www2.mrc-lmb.cam.ac.uk/personal/pemsley/coot/ |
| PHENIX | Afonine et al, 2018 | https://www.phenix-online.org |

| Reagent/resource | Reference or source | Identifier or catalog number |
|---|---|---|
| Chimera X | Meng et al, 2023 | https://www.cgl.ucsf.edu/chimera/ |
| **Other** | | |
| Titan Krios microscope 300 kV | Thermo Fisher | |
| ChemiDoc Imaging System | Bio-Rad, USA | |
| SpectraMax i3x | Molecular Devices, USA | |
| Biacore Sensor Chip CM5 | Cytiva, USA | |
| Biacore 8 K | Cytiva, USA | |
| RX-C18 column | Zhongpu Science, China | |
| Agilent 1260 Infinity II LC System | Agilent, USA | |
| AUTOFLEX III MALDI-TOF | Bruker Corporation, German | |

## Protein expression and purification

The Csm6-2 cDNA (GenBank ID: WP_136192673) was synthesized by GenScript Corporation (Nanjing, China) after codon optimization and cloned into the pET-28a vector, with an N-terminal His$_6$ tag. The pET-28a-Csm6-2 plasmid was transformed into *E. coli* Rosetta (DE3) cells. Protein expression was induced by 0.5 mM isopropyl β-D-1-thiogalactopyranoside (IPTG) overnight at 17 °C. Cells were harvested and resuspended in lysis buffer containing 50 mM Tris-HCl (pH 7.5), 300 mM NaCl, 5% glycerol, and 10 mM imidazole. After sonication and centrifugation, the His$_6$-tagged protein was pooled onto Ni-NTA column (Union Biotech, China). After thorough washing, the protein was eluted with lysis buffer supplemented with 200 mM imidazole. The protein was further purified using 15Q and Superdex 200 10/300 GL columns (GE Healthcare Life Sciences), and stored at −80 °C in 50 mM Tris-HCl (pH 7.5) and 250 mM NaCl. The expression and purification of Csm6-2 mutants followed the same protocol as described above.

## ssRNA cleavage assay

The ssRNA cleavage activities of Csm6-2 and its variants were assessed using both the denaturing polyacrylamide gel electrophoresis (PAGE) and fluorescence resonance energy transfer (FRET)-based assays as previously described (Du et al, 2024). For the gel-based assay, 250 nM FAM-labeled ssRNA (5'-ACUG-CAACGCAAUAUACCAUAGCU-3') was incubated with 100 nM cOA (cA$_4$, cA$_5$ or cA$_6$, Biolog Life Science Institute, Germany) and 1–100 nM Csm6-2 at 37 °C for 45 min, in cleavage buffer containing 20 mM Tris-HCl (pH 7.0), 50 mM KCl, and 25 mM EDTA. Subsequently, the reaction products were separated by a 12% urea denaturing gel, which was imaged using the ChemDoc Touch imaging system (Bio-Rad).

For the FRET-based assay, 200 nM synthetic RNA reporter (FAM-UGUUCGACGA-BHQ1) was incubated with 1 nM Csm6-2 and 10 nM cA$_5$ (or 200 nM Csm6-2 and 200 nM cA$_6$ for cA$_6$ activation assay) in cleavage buffer containing 20 mM Tris-HCl (pH 7.0), 50 mM KCl, and 25 mM EDTA. Fluorescence values were monitored by a microplate reader (SpectraMax i3x) for 1 h at 1-min intervals, with excitation at 490 nm and emission at 520 nm.

To investigate Csm6-2 activation by linear oligonucleotides, we adapted the *Lbu*Cas13a-Csm6 tandem nuclease assay as previously described (Liu et al, 2021). Briefly, 40 nM *Lbu*Cas13a was pre-incubated with 20 nM crRNA in 25 mM HEPES (pH 7.0), 50 mM KCl, 10 mM MgCl$_2$, and 5% glycerol. A mixture containing 100 nM Csm6-2, 200 nM reporter RNA (FAM-10A-BHQ1), and 2 μM activator RNA (A$_{2-6}$U$_6$) was then added. The reaction was initiated by adding 100 pM target RNA complementary to the crRNA spacer region. Fluorescence intensity was monitored using a microplate reader.

## Surface plasmon resonance (SPR) assay

The binding affinity of Csm6-2 to cA$_5$ and cA$_6$ was measured by SPR assay using the Biacore 8 K biosensor instrument (Cytiva, USA). Briefly, 100 μg/mL Csm6-2 was diluted in 10 mM sodium acetate buffer (pH 4.0) and immobilized onto a CM5 sensor chip via the amine coupling method, yielding a final response value of 5148.4 response unit (RU). A multi-cycle kinetics / affinity method was carried out with PBST (137 mM NaCl, 2.7 mM KCl, 1.8 mM KH$_2$PO$_4$, 10 mM Na$_2$HPO$_4$, 0.5% v/v Tween-20) as running buffer. Serial concentrations of cA$_5$ or cA$_6$ were injected onto the chip with binding and dissociation duration of 60 s and 420 s, respectively. After each cycle, the chip surface was regenerated with glycine (pH 3.0). Real-time signals were processed and analyzed using Biacore 8 K evaluation software.

## cOA cleavage assay

The cOA cleavage activity of Csm6-2 was assessed by HPLC and MALDI-TOF MS analyses, as previously described (Du et al, 2023). Briefly, in a 50-μL reaction, 40 μM cOA was incubated with 2 μM Csm6-2 at 37 °C for 2 h, in cleavage buffer containing of 20 mM Tris-HCl (pH 7.0) and 50 mM KCl. Reaction products were extracted using an equal volume of chloroform-isopentanol mixture (24:1), and the aqueous phase was collected for further HPLC and MALDI-TOF MS analysis.

HPLC analysis was conducted on a Agilent 1260 Infinity II LC System, equipped with an RX-C18 column (2.1 × 100 mm, 5 μm) (Zhongpu Science). All components were eluted using a linear gradient of mobile phase A (0.1% trifluoroacetic acid in water) and mobile phase B (0.1% trifluoroacetic acid in acetonitrile) at a column temperature of 40 °C. The eluent was monitored by UV detection at 259 nm.

For MALDI-TOF MS analysis, cOA and its cleavage products were mixed with matrix solution, and 1 μL of the mixture was applied onto a 384 MTP AnchorChip. After drying and crystal formation, the chip was transferred to the excitation source for analysis, using a frequency-tripled Nd:YAG (355 nm) laser. The source region (metal probe) was maintained at 20 kV (AUTOFLEX III MALDI-TOF, Bruker Corporation, Germany). Samples were

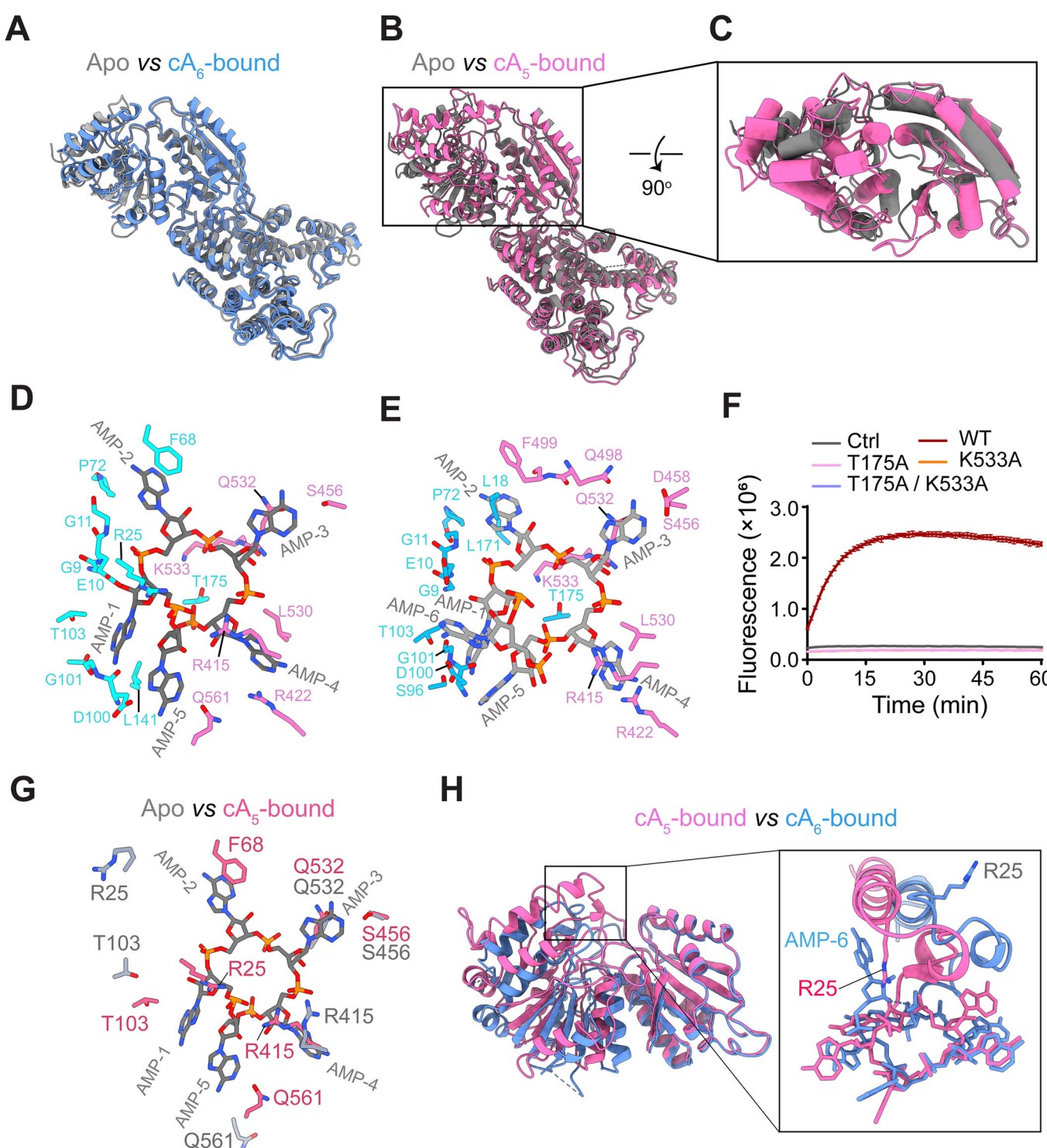

**Figure 4. Mechanism of specific cOA recognition.**

(A) Structural alignment of the $cA_6$-bound Csm6-2 (blue) with its apo form (grey). (B) Structural alignment of the $cA_5$-bound Csm6-2 (pink) with its apo form (grey). (C) Conformational changes in the CARF domain induced by $cA_5$ binding. (D, E) Close-up views of Csm6-2 CARF domain interacting with $cA_5$ (D) and $cA_6$ (E). cOAs molecules are shown as grey sticks. (F) Effect of CARF domain mutations on $cA_5$-dependent activation of Csm6-2 ribonuclease activity, using 1 nM Csm6-2 and 10 nM $cA_5$. Values are means ± SD, $n = 3$ replicates. (G) Conformational shifts in the ligand-binding site upon $cA_5$ binding. (H) Structural alignment of the $cA_5$-bound Csm6-2 CARF domain (pink) with the $cA_6$-bound form (blue). The zoomed-in view on the right highlights the steric hindrance imposed by AMP-6 of $cA_6$ that restricts the movement of R25. Source data are available online for this figure.

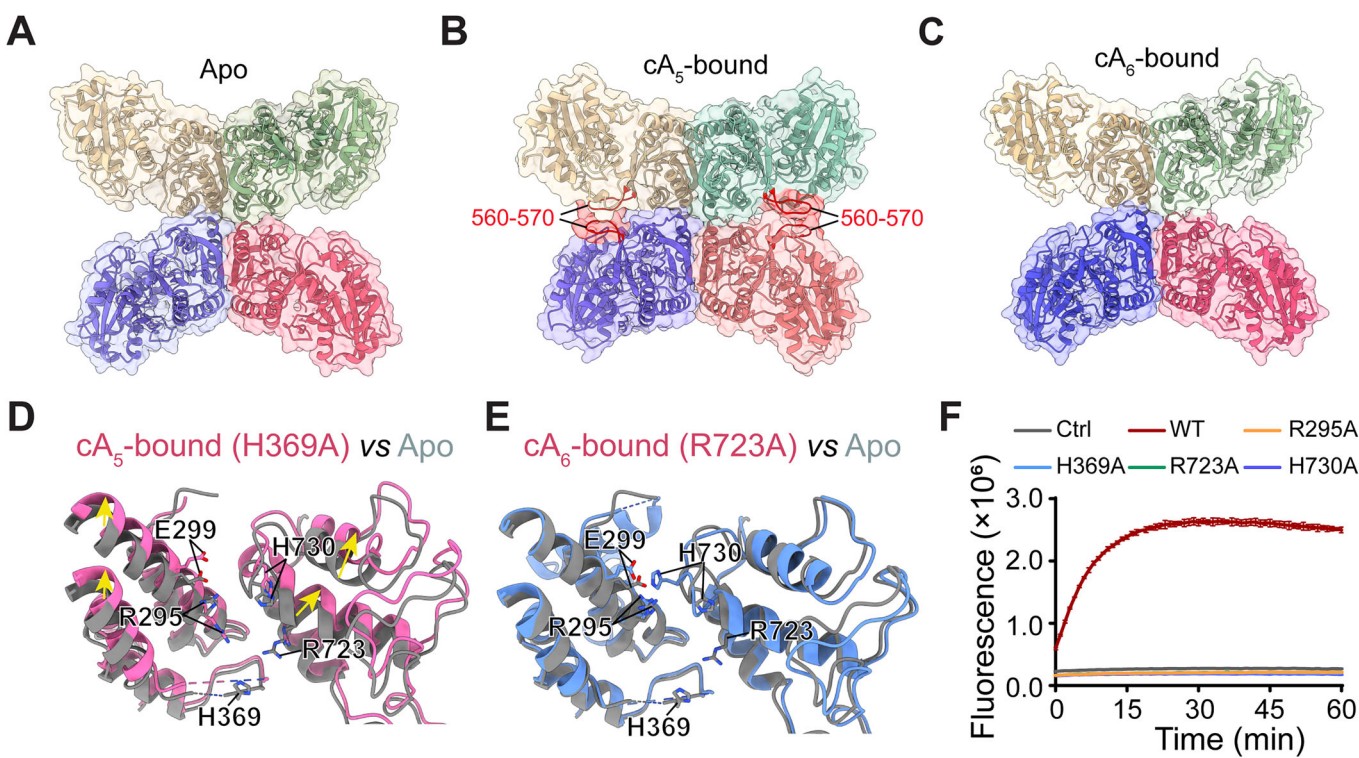

**Figure 5.  Mechanism of allosteric activation.**

(A–C) Comparison of the tetramerization interface mediated by the CARF domain of Csm6-2 in the apo (**A**), $cA_5$-bound (**B**) and $cA_6$-bound (**C**) states. CARF domains from four monomers are shown in distinct colors, overlaid with an 80% transparent surface. Loops 560-570 are highlighted in red. (**D**, **E**) Structural alignment of the HEPN domain in $cA_5$-bound (**D**) and $cA_6$-bound (**E**) Csm6-2 with its apo form (grey). Major conformational changes are indicated by yellow arrows. Catalytic residues are shown as sticks. Disordered regions are indicated by dashed lines. (**F**) Effect of HEPN domain mutations on $cA_5$-dependent activation of Csm6-2 ribonuclease activity, using 1 nM Csm6-2 and 10 nM $cA_5$. Values are means ± SD, $n = 3$ replicates. Source data are available online for this figure.

analyzed using FlexControl software, with data processed via FlexAnalysis.

### Cryo-EM data collection and image processing

For cryo-EM grid preparation, 3 µl Csm6-2 samples (~0.5 mg/ml) were applied onto glow-discharged holey carbon grids (Quantifoil Cu with 2 nm Carbon, R1.2/1.3, 300 mesh), blotted with a Vitrobot Marker IV (Thermo Fisher Scientific) for 3 s under 100% humidity at 4 °C, and subjected to plunge freezing into liquid ethane. Cryo-EM data for Csm6-2 (Apo) was collected using the Thermo Fisher Titan Krios microscope at 300 kV equipped with a Gatan K3 Summit direct electron detector (super-resolution mode, at a nominal magnification of 81,000, pixel size 1.087) and a GIF-quantum energy filter; cryo-EM data for Csm6-2-$cA_6$ and Csm6-2-$cA_5$ were collected using the Thermo Fisher Titan Krios microscope at 300 kV equipped with a Falcon 4i Summit direct electron detector (at a nominal magnification of 130,000, pixel size 0.97) and a GIF-quantum energy filter. Total electron doses were set at 50 e⁻/Å². Defocus values were set from −1.0 to −2.0 µm. EPU (Thermo Fisher) was used for fully automated data collection.

All micrograph stacks were motion corrected with MotionCor2 (Zheng et al, 2017), resulting in a pixel size of 1.087 or 0.97 Å, indicated on the flowchart. Contrast transfer function (CTF)

parameters were estimated using Gctf. Most steps of image processing were performed using cryoSPARC (Punjani et al, 2017).

For 3D processing of apo Csm6-2, 4,304,615 particles were auto-picked from 1608 micrographs. These particles were extracted with Bin 4 and underwent multiple rounds of reference-free 2D classification. After removing obvious ice contaminants and junk particles, 1,484,826 particles were retained and then re-extracted without binning. Next, ab initio models were constructed and employed for heterogeneous 3D refinement. The resulting class of 149,420 particles was re-extracted, further classified via 2D and ab initio methods, and subjected to Non-Uniform reconstruction for subsequent structural analysis. The overall resolution of the apo Csm6-2 map was determined as 2.59 Å using the Fourier Shell Correlation (FSC) 0.143 criterion (Chen et al, 2013).

For 3D processing of the Csm6-2-$cA_6$ dataset, 7,333,325 particles were automatically selected from 5340 micrographs. Extracted with Bin 4, these particles were subjected to multiple rounds of reference-free 2D classification. Following exclusion of ice contaminants and junk particles, 1,672,108 particles were kept and re-extracted without binning. Ab initio models were then generated and used for heterogeneous 3D refinement. The class containing 667,978 particles was subjected to further non-uniform refinement and local refinement (both with and without

D2 symmetry application) for structural analysis. The global resolution of the Csm6-2-cA$_6$ map was 2.53 Å based on the FSC 0.143 criterion.

For 3D processing of the Csm6-2-cA$_5$ data, 1,969,041 particles were auto-picked from 1994 micrographs. After extraction with Bin 4, the particles underwent several rounds of reference-free 2D classification. Post-removal of ice contamination and junk particles, 183,599 particles were retained and re-extracted without binning. Ab initio models were built and subsequently utilized for heterogeneous 3D refinement. The class of 143,784 particles was then processed with further non-uniform refinement and local refinement (with and without D2 symmetry) for structural analysis. The overall resolution of the Csm6-2-cA$_5$ map was 2.67 Å as per the FSC 0.143 criterion.

## Model building

The structure of Csm6-2 monomer predicted by Alphafold 2 (Jumper et al, 2021; Varadi et al, 2024) was used as the starting models and docked into the final EM maps with UCSF Chimera (Pettersen et al, 2004). The models were manually adjusted and iteratively built in COOT (Emsley et al, 2010) and then refined against summed maps using phenix.real_space_refine implemented in PHENIX (Afonine et al, 2018) until the validation data were reasonable. FSC values were calculated between the resulting models and the two half-maps, as well as the averaged map of the two half-maps. The quality of the models was evaluated with MolProbity (Chen et al, 2010). The structure validation statistics were listed in Table 1. All structural figures were prepared with Chimera X (Meng et al, 2023).

## Data availability

The cryo-EM density maps have been deposited to the Electron Microscopy Data Bank under accession numbers EMD-65609 (apo Csm6-2), EMD-65610 (Csm6-2-cA$_6$) and EMD-65611 (Csm6-2-cA$_5$). The corresponding atomic coordinates are available in the RCSB Protein Data Bank (https://www.rcsb.org) under accession codes 9W3U (apo Csm6-2), 9W3V (Csm6-2-cA$_6$) and 9W3W (Csm6-2-cA$_5$).

The source data of this paper are collected in the following database record: biostudies:S-SCDT-10_1038-S44318-026-00767-3.

## Peer review information

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

## Acknowledgements

Single particle cryo-EM data were collected at the Westlake University Cryo-EM Facility. We thank the Westlake University High-Performance Computing Center for computational resources and technical assistance. This work is supported by the National Natural Science Foundation of China (ZL, 32471255; HG, 32271258), and the Natural Science Foundation of Fujian Province (ZL, 2024J02006).

## Author contributions

**Ruyi Shi**: Resources; Data curation; Software; Validation; Investigation; Visualization; Methodology. **Mengquan Yang**: Resources; Data curation; Software; Validation; Investigation; Visualization; Methodology. **Yusong Liu**: Data curation; Validation; Visualization; Methodology. **Haishan Gao**: Resources; Data curation; Software; Supervision; Funding acquisition; Validation; Visualization; Project administration; Writing—review and editing. **Zhonghui Lin**: Conceptualization; Data curation; Formal analysis; Supervision; Funding acquisition; Validation; Visualization; Writing—original draft; Project administration; Writing—review and editing.

Source data underlying figure panels in this paper may have individual authorship assigned. Where available, figure panel/source data authorship is listed in the following database record: biostudies:S-SCDT-10_1038-S44318-026-00767-3.

## Disclosure and competing interests statement

The authors declare no competing interests.

