## [Peer Review File · The EMBO Journal]

Mechanistic Basis for Selective Csm6-2 Activation by Cyclic Penta-Adenylate in a Type III CRISPR-Cas system

Ruyi Shi, Mengquan Yang, Yusong Liu, Haishan Gao, and zhonghui lin

Corresponding author(s): zhonghui lin (zhonghui.lin@fzu.edu.cn) , Haishan Gao (gaohaishan@westlake.edu.cn)

Review Timeline:

Submission Date:	24th Aug 25
Editorial Decision:	25th Nov 25
Revision Received:	20th Jan 26
Editorial Decision:	19th Feb 26
Revision Received:	24th Feb 26
Accepted:	10th Mar 26

Editor: Cornelius Schneider

Transaction Report:

Dear Prof. Lin,

Thank you for submitting your manuscript for consideration by the EMBO Journal. It has now been seen by two referees. A third referee had agreed to review but did not return their report despite several chasers. We have therefore decided to proceed with the reports of two referees which you can find below.

As you can see from the reports both referees think that the findings are interesting and relevant. Referee #1 voices several major concerns which we think are fair and productive.

Given the referees' positive recommendations, I would like to invite you to submit a revised version of the manuscript, addressing the comments of both reviewers. I should add that it is EMBO Journal policy to allow only a single round of revision, and acceptance of your manuscript will therefore depend on the completeness of your responses in this revised version.

When preparing your letter of response to the referees' comments, please bear in mind that this will form part of the Review Process File, and will therefore be available online to the community. For more details on our Transparent Editorial Process, please review our Editorial Policies: <https://link.springer.com/partners/embo-press/editorial-policies>

Thank you for the opportunity to consider your work for publication. I look forward to your revision.

Yours sincerely,

Cornelius Schneider, PhD
Editor
The EMBO Journal
c.schneider@embojournal.org

Read our guidance for manuscript revisions and related editorial policies: <https://link.springer.com/journal/44318/submission-guidelines#cms-Revised-submissions>

<https://media.springernature.com/original/springer-cms/rest/v1/content/27825798/data/v1>

- a point-by-point response to the referees' comments, with a detailed description of the changes made (as a word file).
- a word file of the manuscript text.
- individual production quality figure files (one file per figure)
- a complete author checklist
- Expanded View files (replacing Supplementary Information)
- a Reagents and Tools Table as part of the Methods section

Please remember: Digital image enhancement is acceptable practice, as long as it accurately represents the original data and conforms to community standards. If a figure has been subjected to significant electronic manipulation, this must be noted in the figure legend or in the 'Materials and Methods' section. The editors reserve the right to request original versions of figures and

the original images that were used to assemble the figure.

We realize that it is difficult to revise to a specific deadline. In the interest of protecting the conceptual advance provided by the work, we recommend a revision within 3 months (23rd Feb 2026). Please discuss the revision progress ahead of this time with the editor if you require more time to complete the revisions. Use the link below to submit your revision:

Referee #1:

Shi et al report a structural and biochemical characterisation of an unusual Csm6 crisper effector Csm6-2 (here called ApCsm6). They demonstrate that this is an unusual example of a cA5 activated enzyme, which degrades cA5 using its HEPN domain. As previously reported, cA6 also activates the enzyme, but at higher concentrations. The apo protein is a tetramer, which appears important for activity. CryoEM structures are presented in complex with cA6 and cA5, revealing how nucleotide binding at the CARF domains results in activation of the HEPN ribonuclease, broadly in line with previous studies of Csm6 proteins.

The primary discovery that this enzyme is activated by cA5 is very interesting - expanding our understanding of cOA signalling.

Major points:

1. The EC50 value of 0.09 nM for cA5 binding to Csm6-2 suggests extremely tight binding and is unprecedented in the literature for nucleotides binding to proteins. If the authors wish to claim such tight binding in this case, further data is required to support the claim. Notably, the raw data underlying figure 1b has not been provided and the method used to generate the EC50 data is not described sufficiently.
2. In figure 1d, why are there two peaks for cA5? What does "aggregated cA5" mean? Why is only one peak observed with cA6? Aggregation of nucleotides is not a recognised phenomenon.
3. Please include more information on the details of the assays used to check all the variant proteins reported in the paper - how was a single fluorescence value determined for each variant and how is this justified - why not measure a rate of reaction rather than an end point?
4. The L473/474E variant has an unusual behaviour on the size exclusion column (figure 3), with multiple overlapping peaks. This should be discussed - are all the peaks due to variant protein? Is the protein stably folded?
5. Why was the H369A variant used for the cA5 structure and the R723A variant used for the cA6 structure?
6. Considering the structures with cA5 and cA6, given that cA6 is an activator at higher concentrations, it presumably allows Csm6-2 to access the same activated conformation as cA5. How do the authors explain this, given the marked differences seen between the two structures?
7. The mutagenesis of the cA6 binding site (T175 and K533) is not described in adequate detail. How were the variant proteins assayed? Were they checked using both cA5 and cA6? What concentration of activator was used and was activity concentration dependent - as seen for the wt in figure 1b? As written, the manuscript implies this is cA6-activated enzyme as it's shown with the cA6 bound structure, but is this correct?
8. The statement (P12) that tetramerization is essential for activity is not supported by the data showing that the L473/4E variant still has activity. Please revise the text.
9. The observation that cA5 is the activator of this protein is very interesting. In the discussion, it would be worth mentioning that this may be a cellular response to avoid viral ring nucleases. A similar phenomenon has been observed in the type III crisper systems that utilise SAM-AMP rather than cOA signalling.

Minor points:

1. The enzyme studied here has already been named as Csm6-2. Why rename it? Given its significantly unusual properties, it should probably have a name that discriminates it from standard Csm6 proteins.
2. Abstract typo relating should be relying
3. Introduction first para: although type III systems detect viral RNA, they have many ways to combat viral infection that don't directly defend against viral RNA - consider rephrasing.
4. P4. Some Csm6 family proteins referenced here degrade their activator using both the CARF and HEPN domains - please rephrase.
5. P5. SAVED domains are also found in CRISPR associated effectors, such as CalpL - please add here.
6. P5 para 1. Typo "presents" should be "is present".
7. P5 para 2. Given point 3, Csm6-2 is not unique in degrading cOA via its HEPN domain - please rephrase. The unique aspect of Csm6-2 is the use of a cA5 activator - this could be emphasised more strongly in the final paragraph here.
8. Results para 1. Please state here that Csm6-2 was previously shown to be activated by cA6 but not cA3 or cA4 (ref 36).
9. Page 8. L473 and 474 were mutated to glutamate, not glutamine.
10. P12. Most ring nucleases are likely not "acquired viral" enzymes but cellular enzymes that have been acquired by viruses. Please rephrase.

Referee #2:

The manuscript by Shi et al., reported the first cA5 activated effector ApCsm6 from type III CRISPR-system, and discovered that ApCsm6 degrade cOAs via its HEPN domain. By determining the Cryo-EM structures of ApCsm6 and its complexes with cA5 and cA6, they also elucidated the activation mechanism. The findings reported in this manuscript is quite novel and advanced our understanding of type III CRISPR-system. I recommend publication of this work after minor revision.

Minor points.

1. The authors concluded that ApCsm6 self-limits its ribonuclease activity by degrading cOAs via its HEPN domain, however, it seemed that the cA5 cleaved product A5>P is still quite active. Can you make a side by side comparison using cA5 and A5>P?
2. In the abstract, the authors mentioned that "Although cOA-activated effectors have been extensively characterized, the cA5-specific effectors remained unexplored despite cA5 being among the most abundant cOA species produced during phage infection." I am not sure if this statement is correct or not. And it is better to describe it in the introduction as well and cite the reference.

A point-by-point response to the referees' comments

Referee #1:

Shi et al report a structural and biochemical characterisation of an unusual Csm6 crisper effector Csm6-2 (here called ApCsm6). They demonstrate that this is an unusual example of a cA5 activated enzyme, which degrades cA5 using its HEPN domain. As previously reported, cA6 also activates the enzyme, but at higher concentrations. The apo protein is a tetramer, which appears important for activity. CryoEM structures are presented in complex with cA6 and cA5, revealing how nucleotide binding at the CARF domains results in activation of the HEPN ribonuclease, broadly in line with previous studies of Csm6 proteins. The primary discovery that this enzyme is activated by cA5 is very interesting - expanding our understanding of cOA signalling.

Response: We are grateful to the reviewer for accurately summarizing the main findings of our study and for the encouraging comments on the manuscript.

Major points:

1. *The EC50 value of 0.09 nM for cA5 binding to Csm6-2 suggests extremely tight binding and is unprecedented in the literature for nucleotides binding to proteins. If the authors wish to claim such tight binding in this case, further data is required to support the claim. Notably, the raw data underlying figure 1b has not been provided and the method used to generate the EC50 data is not described sufficiently.*

Response: Thank you for raising this important point. In response, we have directly measured the binding affinity between Csm6-2 and cOA activators using surface plasmon resonance (SPR). The SPR data revealed that cA5 and cA6 bind Csm6-2 with comparable affinity, exhibiting K_D values of 1.41 nM and 3.72 nM, respectively (Fig. 1C and Fig. S1A). This suggests that the marked difference in their ability to activate ribonuclease activity cannot be attributed primarily to differences in binding affinity.

To further support this, we performed cleavage assays with a saturating concentration of cOAs (100 nM) (Fig. 1A). In the presence of cA4, Csm6-2 showed no detectable activity. With cA6, weak activity was observed only at 100 nM enzyme concentration. In contrast, robust ssRNA cleavage was triggered by cA5 at an enzyme concentration of 1 nM. These functional results demonstrate that the striking difference in activation between cA5 and cA6 stems primarily from divergent allosteric effects rather than differential binding. This is consistent with our structural observations that cA5 binding induces large conformational changes in both the CARF and HEPN domains, while the cA6-bound structure remains nearly identical to the apo state.

We have incorporated these new results and the corresponding discussion into the revised manuscript. In addition, as suggested by the reviewer, the original EC₅₀ data in Fig. 1b have been replaced with the direct binding data from our SPR experiments to accurately represent the interaction between Csm6-2 and cOAs. All relevant raw data, as well as a comprehensive description of the experimental methods, are now provided in the revised manuscript and supplementary materials.

2. In figure 1d, why are there two peaks for cA5? What does "aggregated cA5" mean? Why is only one peak observed with cA6? Aggregation of nucleotides is not a recognised phenomenon.

Response: We thank the reviewer for raising this important point. To resolve this issue, we diluted the cA5 reaction mixture with distilled water at a 1:1 (v/v) ratio and reanalyzed it using HPLC under the same conditions as in the original experiment. Under these diluted conditions, only a single peak was observed (Fig. 1D), suggesting that cA5 may form higher-order multimers or self-associated complexes at high concentrations. Interestingly, such aggregation behavior was not observed for cA6 under identical conditions, possibly due to differences in molecular structure or physicochemical properties, though the exact reason remains unclear. We have updated the data in Fig. 1D of the revised manuscript.

3. Please include more information on the details of the assays used to check all the variant proteins reported in the paper - how was a single fluorescence value determined for each variant and how is this justified - why not measure a rate of reaction rather than an end point?

Response: We thank the reviewer for the valuable suggestions. In response, we have re-evaluated the ssRNA cleavage activity of the Csm6-2 mutants using a kinetic assay mode rather than a single end-point measurement. We have updated the data in Fig. 4F and Fig. 5F, and provided more information on the details of the assays both in the Methods section and in the corresponding figure legends.

4. The L473/474E variant has an unusual behaviour on the size exclusion column (figure 3), with multiple overlapping peaks. This should be discussed - are all the peaks due to variant protein? Is the protein stably folded?

Response: Thank you for highlighting this important observation. In response, we have re-purified the protein of L473/474E variant and analyzed it again by size exclusion chromatography. The protein now elutes as a single, symmetrical peak corresponding to a dimeric species, indicating that it is properly folded and monodisperse under the optimized conditions. The multiple overlapping peaks observed in the initial experiment, which suggested the presence of aggregates, likely resulted from suboptimal handling or instability during the earlier preparation. We have updated the data in Fig. 2H-I of the revised manuscript.

5. *Why was the H369A variant used for the cA5 structure and the R723A variant used for the cA6 structure?*

Response: As shown in Fig. 1D – E, Csm6-2 degrades cOA activators via its HEPN domain. The HEPN ribonuclease active site consists of four catalytic residues: R295, H369, R723, and H730. Alanine substitution of each residue completely abolished ssRNA cleavage (Fig. 5F), indicating that any of these mutations can prevent cOA degradation. For structural analysis, we used two catalytically deficient mutants, H369A and R723A, to prepare the cA5-bound and cA6-bound complexes, respectively. Using two different active-site mutants allowed us to examine potential conformational changes at each catalytic residue.

6. *Considering the structures with cA5 and cA6, given that cA6 is an activator at higher concentrations, it presumably allows Csm6-2 to access the same activated conformation as cA5. How do the authors explain this, given the marked differences seen between the two structures?*

Response: We thank the reviewer for this insightful question. Although cA₆ and cA₅ binds Csm6-2 with comparable affinity, only cA₅ induces CARF domain closure, enhances global tetramer stabilization, and remodels the active site in the HEPN domain. Structural analyses reveal that the sixth AMP of cA₆ imposes significant steric hindrance on CARF domain movement, thereby preventing its closure and subsequent allosteric activation (Fig. 4H). Furthermore, ssRNA cleavage assays showed that at a saturating cOA concentration (100 nM), cA₅ triggers robust cleavage with only 1 nM enzyme, whereas cA₆ elicits only weak activity even at 100 nM enzyme (Fig. 1A). These findings indicate that the cA₅-bound conformation represents a fully activated state, while cA₆ induces a suboptimal, weakly active conformation.

To clarify these differences, we have reorganized the structural and biochemical data in Figures 3~5 to enable direct comparison between the cA₅- and cA₆-bound states. We hope this presentation elucidates the mechanistic basis for the distinct activation profiles of the two ligands.

7. *The mutagenesis of the cA6 binding site (T175 and K533) is not described in adequate detail. How were the variant proteins assayed? Were they checked using both cA5 and cA6? What concentration of activator was used and was activity concentration dependent - as seen for the wt in figure 1b? As written, the manuscript implies this is cA6-activated enzyme as it's shown with the cA6 bound structure, but is this correct?*

Response: We apologize for the insufficient description. The ribonuclease activity of the T175A and K533A mutants was assessed using the FRET-based ssRNA cleavage assay (see Methods). Both mutants were tested for activation by cA₅ and cA₆. To account for the significantly weaker activation by cA₆ relative to cA₅, the assay for cA₆ employed 200nM Csm6-2 and 200nM cA₆, whereas the cA₅ assay used 1nM

Csm6-2 and 10nM cA5. Additional details of the assays are now provided in the Methods section and in the corresponding figure legends.

8. *The statement (P12) that tetramerization is essential for activity is not supported by the data showing that the L473/4E variant still has activity. Please revise the text.*

Response: We thank the reviewer for highlighting this important point. The text on page 12 has been revised to: "Since tetramerization is critical for Csm6-2's ribonuclease activity, these additional intermonomer contacts upon may stabilize the catalytically active tetramer." (page 13 of the revised manuscript).

9. *The observation that cA5 is the activator of this protein is very interesting. In the discussion, it would be worth mentioning that this may be a cellular response to avoid viral ring nucleases. A similar phenomenon has been observed in the type III crisper systems that utilise SAM-AMP rather than cOA signalling.*

Response: We appreciate the reviewer for this insightful suggestion. We have included this discussion in the revised manuscript.

Minor points:

1. *The enzyme studied here has already been named as Csm6-2. Why rename it? Given its significantly unusual properties, it should probably have a name that discriminates it from standard Csm6 proteins.*

Response: We have changed the name of the protein to 'Csm6-2' throughout the manuscript.

2. Abstract typo relaying should be relying.

Response: Yes, it should be relying.

3. *Introduction first para: although type III systems detect viral RNA, they have many ways to combat viral infection that don't directly defend against viral RNA - consider rephrasing.*

Response: We thank the reviewer for this point have rephrased the text accordingly.

4. *P4. Some Csm6 family proteins referenced here degrade their activator using both the CARF and HEPN domains - please rephrase.*

Response: Thank you for pointing out this. We have included this at the end of the second paragraph in Introduction.

5. *P5. SAVED domains are also found in CRISPR associated effectors, such as CalpL - please add here.*

Response: We have added CalpL as an additional example that binds cOA with SAVED domain at the end of the 3rd paragraph in Introduction.

6. P5 para 1. Typo "presents" should be "is present".

Response: The typo has been corrected. Thanks!

7. P5 para 2. Given point 3, Csm6-2 is not unique in degrading cOA via its HEPN domain - please rephrase. The unique aspect of Csm6-2 is the use of a cA5 activator - this could be emphasised more strongly in the final paragraph here.

Response: Thanks for the suggestion. We have rephrased the statement accordingly.

8. Results para 1. Please state here that Csm6-2 was previously shown to be activated by cA6 but not cA3 or cA4 (ref 36).

Response: We have added this statement in the first sentence of Results para 1.

9. Page 8. L473 and 474 were mutated to glutamate, not glutamine.

Response: The typo has been corrected. Thanks!

10. P12. Most ring nucleases are likely not "acquired viral" enzymes but cellular enzymes that have been acquired by viruses. Please rephrase.

Response: The statement has been revised accordingly.

Referee #2:

The manuscript by Shi et al., reported the first cA5 activated effector ApCsm6 from type III CRISPR-system, and discovered that ApCsm6 degrade cOAs via its HEPN domain. By determining the Cryo-EM structures of ApCsm6 and its complexes with cA5 and cA6, they also elucidated the activation mechanism. The findings reported in this manuscript is quite novel and advanced our understanding of type III CRISPR-system. I recommend publication of this work after minor revision.

Response: We appreciate the reviewer for the positive and encouraging comments on our manuscript. We have performed additional experiments and revised part of the statements to address the reviewer's comments. As a result, the paper has been greatly improved.

Minor points.

1. *The authors concluded that ApCsm6 self-limits its ribonuclease activity by degrading cOAs via its HEPN domain, however, it seemed that the cA5 cleaved product A5>P is still quite active. Can you make a side by side comparison using cA5 and A5>P?*

Response: We thank the reviewer for raising this insightful point. In response, we have performed a side by side comparison between cA5 and A5>P. The results below

demonstrated that while both cA5 and A5>P are active in ApCsm6 activation, cA5 is more effective than A5>P. This finding aligns with previous reports showing that, in addition to cyclic tetraadenylate (cA4), Csm6 can also be activated by the linear tetra-adenylate terminated with a 2',3'-cyclic phosphate (A4>P) (Science 357, 605–609 (2017); Nature 548, 543–548 (2017)). Therefore, to characterize the cOA activation profiles of ApCsm6, we employed the collateral nuclease activity of Cas13a to produce A_n>P activators, as previously described (Gootenberg *et al*, 2018).

2. In the abstract, the authors mentioned that "Although cOA-activated effectors have been extensively characterized, the cA5-specific effectors remained unexplored despite cA5 being among the most abundant cOA species produced during phage infection." I am not sure if this statement is correct or not. And it is better to describe it in the introduction as well and cite the reference.

Response: We thank the reviewer for this important comment. We have rephrased the statement to "..... despite cA5 being one of the most abundant cOA species produced during phage infection". Previously, Smalakyte *et al.* provided direct evidence that cA5 is one of the most abundant cyclic oligoadenylate species produced during phage infection (*Nucleic Acids Res.* 2020, 48, 9204). As shown in Fig. 2A of the reference, HPLC–MS analysis revealed significant accumulation of cA5 (along with cA6) in *E. coli* cells upon MS2 phage infection. We have cited this reference in the Discussion para 1 .

Dear Prof. Lin,

Thank you for submitting a revised version of your manuscript. Your study has now been seen by the original referee #1, who finds that their previous concerns have been addressed and now recommend publication of the manuscript. There remain only a few mainly editorial points that have to be addressed before I can extend formal acceptance of the manuscript:

- As we are switching from a free-text author contribution statement towards a more formal statement based on Contributor Role Taxonomy (CRediT) terms, please remove the present Author Contribution section and instead specify each author's contribution(s) directly in the Author Information page of our submission system during upload of the final manuscript. See <https://casrai.org/credit/> for more information.
- Please adjust the in-text callouts for individual figures and figure panels: e.g. Fig. 2HIJ; callouts for the Appendix Figures need correction - "Appendix" word missing in each callout
- Please provide the APPENDIX FILE WITH ToC as PDF
- Please provide the Reagent and Tools Table. For more information, please check <https://media.springernature.com/original/springer-cms/rest/v1/content/27825802/data/v1>
- SOURCE DATA: Please upload the SD folders separately (one folder per figure)
- Please provide a legend for each movie in a text file and then each movie should be zipped up with its legend and uplidd separately
- Please rename the Methods section into Materials and Methods.
- Please note that the specific URLs for EMD-65609, EMD-65610, EMD-65611, 9W3U, 9W3V, and 9W3W datasets are not provided in the data availability statement.

With best regards,

Cornelius Schneider

Cornelius Schneider, PhD
Editor | The EMBO Journal
c.schneider@embojournal.org

Please refer to our figure preparation guideline in order to ensure proper formatting and readability in print as well as on screen:

<https://link.springer.com/journal/44318/submission-guidelines#cms-Figure-and-data-presentation>

Use the link below to submit your revision:

Referee #1:

The authors have responded constructively to my comments, provided further new data and revised the discussion. I'd be happy to see this published without further delay.

Response to Editorial Points

1. *As we are switching from a free-text author contribution statement towards a more formal statement based on Contributor Role Taxonomy (CRediT) terms, please remove the present Author Contribution section and instead specify each author's contribution(s) directly in the Author Information page of our submission system during upload of the final manuscript. See <https://casrai.org/credit/> for more information.*

Response: We have removed the “Author Contribution” section and specified each author’s contribution in the Author Information page of our submission system.

2. *Please adjust the in-text callouts for individual figures and figure panels: e.g. Fig. 2HIJ; callouts for the Appendix Figures need correction - "Appendix" word missing in each callout.*

Response: The in-text callouts for individual figures and figure panels as well as appendix figures have been adjusted accordingly.

3. *Please provide the APPENDIX FILE WITH ToC as PDF*

Response: The APPENDIX FILE WITH ToC is now provided as PDF.

4. *Please provide the Reagent and Tools Table. For more information, please check <https://media.springernature.com/original/springer-cms/rest/v1/content/27825802/data/v1>*

Response: The Reagent and Tools Table is provided below in accordance with the specified guidelines.

5. *SOURCE DATA: Please upload the SD folders separately (one folder per figure)*

Response: The SD folders have been uploaded separately, with one folder per figure.

6. *Please provide a legend for each movie in a text file and then each movie should be zipped up with its legend and upldd separately.*

Response: Each movie has been zipped together with its corresponding legend (provided in a text file) and uploaded separately as requested.

7. *Please rename the Methods section into Materials and Methods.*

Response: The Methods section has been renamed into Materials and Methods.

8. *Please note that the specific URLs for EMD-65609, EMD-65610, EMD-65611, 9W3U, 9W3V, and 9W3W datasets are not provided in the data availability statement.*

Response: The URLs for each dataset have been provided in the data availability statement.

Dear Prof. Lin,

I am pleased to inform you that your manuscript has been accepted for publication in the EMBO Journal.

You may qualify for financial assistance for your publication charges - either via a Springer Nature fully open access agreement or an EMBO initiative. Check your eligibility: <https://link.springer.com/journal/44318/how-to-publish-with-us>

Yours sincerely,

Cornelius Schneider, PhD
Editor
The EMBO Journal
c.schneider@embojournal.org

Please note that it is The EMBO Journal policy for the transcript of the editorial process (containing referee reports and your response letters) to be published as an online supplement to each paper. If you should prefer removal of any referee-only figures included in the point-by-point response(s), e.g. because they may still be used for future publication or because they have been reproduced from published work by others, please do let us know immediately via response email.

More information is available here: <https://link.springer.com/partners/embo-press/editorial-policies#Peer%20review>
